# Transfer learning for KiTS21 Challenge

No Author Given

No Institute Given

**Abstract.** Transfer learning has witnessed a recent surge of interest after proving successful in multiple applications. However, it highly relies on the quantity of annotated data. Constrained by the labor cost and expertise, it is hard to annotate sufficient organs and tumors at voxel level for medical image segmentation. Consequently, most benchmark datasets were collected for the segmentation of only one type of organs and/or tumors, and all task-irrelevant organs and tumors were annotated as the background. We aim to make use of these partially but plentifully labeled datasets to boost the segmentation performance of annotation-limited KiTS21 segmentation task. To this end, we first construct a general medical image segmentation model that learns to segment these partially labeled organs or tumors. Then we transfer its pretrained weights to a specific downstream task, i.e., KiTS21. The primary experiments demonstrate the effectiveness of the proposed transfer learning strategy.

**Keywords:** Transfer learning · Limited annotation · Kidney tumor segmentation

## 1  Introduction

Automatic kidney tumor segmentation in computed tomography images is one of the most important tasks in the computer-aided diagnosis of kidney diseases. Although deep learning has achieved great success in many medical applications, kidney tumor segmentation still remains challenging due to its limited annotations, which is a common issue for the most of medical image segmentation tasks. Fortunately, there are more and more open-source benchmarks available for the development of medical image segmentation algorithms. However, most of them suffers from the partially labeled issue due to the intensive cost of annotations. To address this issue, zhang *et al* [7] proposed a dynamic on-demand network (DoDNet) that learns to segment multiple organs and tumors by using partially labeled datasets. This makes it more convenient to learn a single segmentation network from the diverse labeled datasets. In this paper, we attempt to transfer the weights pre-trained on partially labeled datasets to downstream task. We conduct experiments on the KiTS21 dataset. The primary results have demonstrated the effectiveness of the proposed transfer learning strategy.

## 2   Methods

Our Method is heavily based on DoDNet [7] and nnUNet [5], the pipeline consist of two part: first, we use dynamic head pre-train our backbone on Multi-Organ and Tumor Segmentation (MOTS) [7] dataset, then transfer the pre-trained weight on KiTS21 task, we illustrate the structure of our model in Fig.1. In the downstream task, we don't use dynamic filter generating and replace dynamic head with a convolution layer.

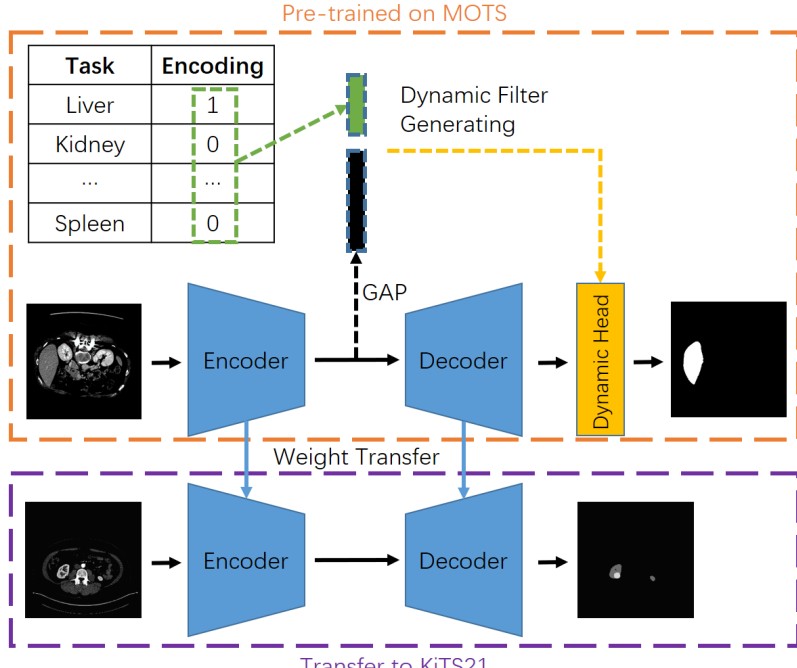

**Fig. 1.** Pipeline of our proposed method. We use dynamic head pretrain a segmentation network on several partially labeled datasets, and then transfer weight to KiTS21 task.

### 2.1   Training and Validation Data

MOTS is composed of seven partially labeled sub-datasets, involving seven organ and tumor segmentation tasks (including LiTS19 [1], KiTS19 [4], and Medical Segmentation Decathlon [6]). There are 1155 3D abdominal CT scans collected from various clinical sites around the world, including 920 scans for training and 235 for test.

Because KiTS21 dataset contain KiTS19 dataset, so MOTS has overlapped with KiTS21, therefore, we only use 210 cases have been used in MOTS pretrain for fine-tune, according to MOTS, we choose 168 images for training and

42 for validation. Notice, our final submission model are fine-tuned on all official KiTS21 training set.

## 2.2    Preprocessing

Our pre-processing strategy is following nnUNet [5], we resample all cases to a common voxel spacing of $0.78126 \times 0.78125 \times 0.78125$, and train the network with a patch size $128 \times 128 \times 128$. The data augmentation methods include scaling, rotations, brightness, contrast, gamma and Gaussian noise augmentations.

## 2.3    Proposed Method

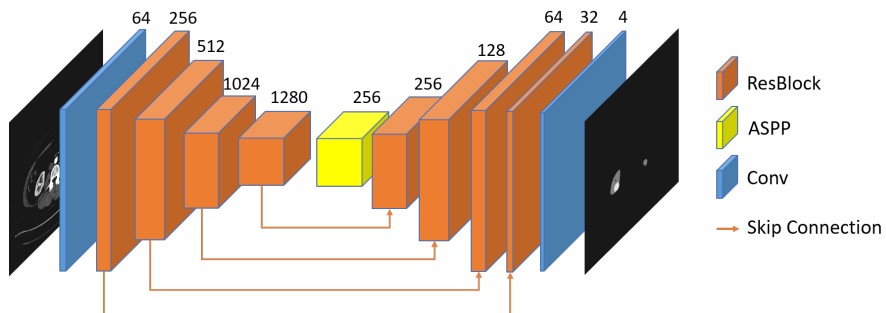

**Fig. 2.** Detailed network architecture, number on the blocks represent the channel size of the outputs.

**Network Architecture** The main component of our framework is Residual 3D U-Net. It use 3D convolutions, LeakyReLU nonlinearities and instance normalization. Upsampling is performed via transposed convolution and downsampling is performed with strided convolutions. The residual blocks of encoder are composed of Conv-instnorm-Conv-instnorm-Conv-instnorm-LeakyReLU. As shown in Fig.2, the encoder have 4 stages, in each stage, we perform downsample at the first residual block, then repeat this basic residual blocks (without downsample) 2, 3, 5 and 2 times, respectively. Inspire by [2], we use ASPP to capture objects as well as useful image context at multiple scales. Different from encoder, The residual blocks of decoder are composed of Conv-instnorm-LeakyReLU-Conv-instnorm, which are similar to [3]. These residual blocks implemented in every stages of decoder only once.

**Loss Function** We train the model with the combination of dice loss and cross entropy loss. For the two Hierarchical Evaluation Classes (HECs) **Kidney and Masses** and **Kidney Mass**, we design a HECs-based cross entropy loss to optimize it. We consider HECs as the foreground and the rest as the background, then calculating the cross entropy loss for **Kidney and Masses** and **Kidney Mass** respectively. Finally, these two kind of HECs-based cross entropy loss was multiplied by the weights of 0.1 and 0.3 then added to the original cross entropy loss.

**Strategy** The stochastic gradient descent (SGD) algorithm with a momentum of 0.99 was adopted as the optimizer. In order to reduce the time consumed in the ablation experiment, all result we reported are training 100 epochs using the nnUNet framework. The learning rate was initialized to 0.01 and decayed according to a polynomial policy $lr = lr_{init} \times (1 - \frac{k}{K})^{0.9}$, where the maximum epoch K was set to 100. Our final submitted model will use 5-fold cross-validation and train 1000 epochs.

## 3   Results

**Table 1.** Performance of different methods. 'SD' means Surface Dice. Kidney, masses and tumor represent HECs **Kidney and Masses**, **Kidney Mass** and **Tumor**, respectively. Notice, all methods are training on nnUNet framework with 100 epochs.

|         | Dice_kidney | Dice_masses | Dice_tumor | SD_kidney | SD_masses | SD_tumor |
|---------|-------------|-------------|------------|-----------|-----------|----------|
| nnUNet  | 0.9405      | 0.7454      | 0.7162     | 0.8898    | 0.6026    | 0.5821   |
| Ours    | **0.9513**  | **0.7953**  | **0.7663** | **0.9178** | **0.6901** | **0.6592** |

We use KiTS21's official code to generate 'groups' of sampled segmentation and evaluate our predictions. The volumetric Dice coefficient and the Surface Dice are uses for evaluation. Table 1 demonstrated the superior performance of our method over nnUNet baseline, and some examples of our prediction results is depicted in Fig. 3.

## 4   Discussion and Conclusion

In this paper, we described a two-stage semantic segmentation pipeline for kidney and tumor segmentation by using dynamic filter generating and multiple partially labeled datasets pre-train. Experiment results demonstrated the value of DoDNet and the MOTS dataset by successfully transferring the weights pre-trained on MOTS to KiTS21 tasks. It suggests that the a pre-trained 3D network is conducive to other small-sample 3D medical image segmentation tasks.

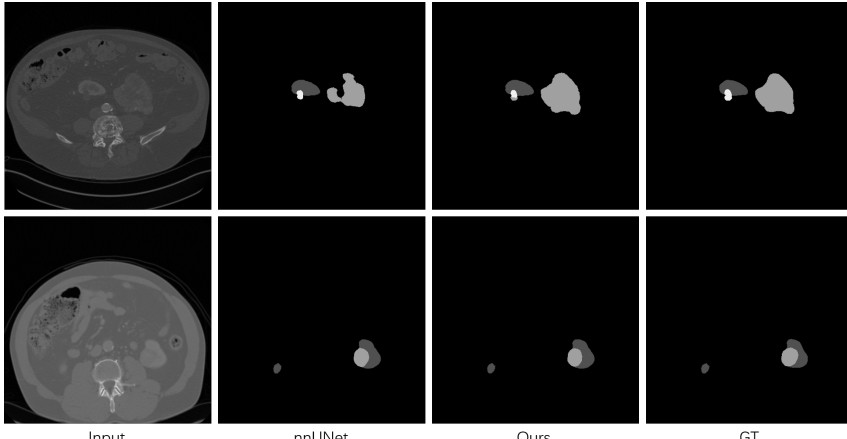

| Input | nnUNet | Ours | GT |

**Fig. 3.** Visualization of segmentation results of case 151 (the first row) and 175 (the second row)

## Acknowledgment

(optional) Feel free to acknowledge funding sources as well as anything or anyone else that you feel deserves an acknowledgment.

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
