# OpenReview forum: "Transfer learning for KiTS21 Challenge"
_MICCAI.org/2021/Challenge/KiTS — Submitted to KiTS21 Challenge_

### Official Review · Reviewer_gxqU · 2021-08-30

**Rating:** 7

**Review:**

The authors present a interesting approach in which they pre-trained their models on a combination of other publicly availble medical imaging datasets and then fine tuned them on the KiTS21 data. However, they did not explain how they synthesized KiTS21's multiple annotations per case during training. Please be sure to add this -- did you use majority voting? Or did you sample segmentations like you did during evaluation?

---

### Official Review · Reviewer_fZvS · 2021-08-30

**Rating:** 6

**Review:**

### Overall

- An interesting and potentially very promising approach
- The "partially-but-plentifully labeled" datasets are unavoidable in this field, and it would be great to be able to make use of them together

### Introduction

- Midway through "most of them suffers" -> "most of them suffer"

### Methods

- "the pipeline consist" -> "the pipeline consists"
- Very nice figure
- "dataset contain KiTS19" -> "dataset contains KiTS19"
- Could you please explain how you chose weights of 0.1 and 0.3 for your two components of the HEC loss function?

### Results

- Please clarify - is nnU-Net in the table the same architecture as "Ours" but just with a different training strategy? If not, do you have results of your full approach but without transfer learning to compare to? If so it might be nice to include it in the table.
- Please update this section with the official results once they are available

### Discussion and Conclusion

- Please remove the acknowledgements section if you don't plan to use it

---

### Decision · Program_Chairs · 2021-08-30

**Decision:**

Major Revisions

**Comment:**

Please address the reviewer comments and resubmit